# Impact of Human Papillomavirus Vaccination on Male Disease: A Systematic Review

**DOI:** 10.3390/vaccines11061083

**Published:** 2023-06-09

**Authors:** Catarina Rosado, Ângela Rita Fernandes, Acácio Gonçalves Rodrigues, Carmen Lisboa

**Affiliations:** 1Division of Microbiology, Department of Pathology, Faculty of Medicine, University of Porto, 4200-039 Porto, Portugal; up201708379@edu.med.up.pt (C.R.); arfernandes@med.up.pt (Â.R.F.); agr@med.up.pt (A.G.R.); 2CINTESIS@RISE, Center of Health Technology and Services Research/Rede de Investigação em Saúde, Faculty of Medicine, University of Porto, 4200-319 Porto, Portugal; 3Department of Dermatology and Venereology, Centro Hospitalar Universitário São João, 4200-319 Porto, Portugal

**Keywords:** human papillomavirus, HPV-related disease, anogenital condyloma, penile intraepithelial neoplasia, anal intraepithelial neoplasia, penile cancer, anal cancer, HPV vaccination, male, men

## Abstract

Human papillomavirus (HPV)-related diseases are highly prevalent in men worldwide, comprising external anogenital condyloma, anal intraepithelial neoplasia (AIN), penile intraepithelial neoplasia (PIN), and anogenital and oropharyngeal cancers. There is exceptionally low vaccine coverage in the male population. Only 4% of men were fully vaccinated, worldwide, as of 2019. The aim of this review is to assess the impact of HPV vaccination on male disease. Three databases (MEDLINE, Web of Science, Scopus) and Clinical Trials.gov were searched. We included thirteen studies, eight randomized controlled trials (RCTs), and five cohorts, comprising a total of 14,239 participants. Regarding anal disease, seven studies reported HPV vaccine efficacy ranging from 91.1% to 93.1% against AIN1, and ranging from 89.6% to 91.7% against AIN2|3 and anal cancer. Five studies showed an efficacy against genital condyloma of 89.9% in HPV-naïve males, varying between 66.7% and 67.2% in intention-to-treat populations. Studies reporting no efficacy have included older participants. These results support vaccination of young men previously infected, beyond HPV-naïve males. The evidence quality was moderate to low for most outcomes, namely genital diseases. RCTs are needed to assess the efficacy of HPV vaccination on male oropharyngeal cancer.

## 1. Introduction

Human papillomavirus (HPV) is the most common agent of sexually transmitted diseases [1]. HPV-related diseases have a high prevalence in men of all ages around the world [2]. Overall, 45 genotypes can specifically invade the anogenital tract and epithelium of the oral cavity, oropharynx, and larynx [3]. The most common genotypes in the male population are HPV-6, 11, 16, and 18 [2]. Benign lesions such as external anogenital warts and recurrent respiratory papillomatosis are caused mostly by low-risk types such as HPV-6 and HPV-11 [4,5]. Malignant anogenital and head and neck diseases are more associated with high-risk (HR)-HPV types, of which HPV-16 and HPV-18 are the most common [2]. HPV infection can resolve naturally without causing disease [6], and even in persistent infection, a large number of infected men do not present clinical manifestations [7]. However, when present, they may range from benign lesions such as anogenital warts or respiratory papillomatosis to more severe situations such as anal and genital cancers and their precursors, and head and neck cancer (HNC) [8]. Since the HPV vaccination of girls and women has shown a protective effect against HPV-related diseases [9], extending the vaccination to boys and men could lead to a lower burden of diseases.

Anogenital warts account for a high burden of HPV-related diseases, with an incidence rate among males of 103 to 168 per 100,000 per year [10]. Recurrent respiratory papillomatosis (RRP) is less frequent [11], being more common in young children, adolescents, and some adults [12]. This disease is characterized by many recurrences in its course, and rarely concludes in squamous cell carcinoma [13]. Regarding HNC, around 70% are estimated to be HPV-attributable, and are less influenced by tobacco and/or drinking [14].

We have been witnessing an alarming increase in the number of anal cancer cases, with an estimated rise of 2% per year [15]. Anal cancer and its precursor lesions (low-grade and high-grade squamous intraepithelial lesions) are the main HPV-related diseases that occur in the anal portion of the digestive tract [16]. The most frequent genotypes associated with anal diseases caused by HPV are HPV-16 and HPV-18, accounting for 70% of the cases.

Men who have sex with men (MSM) are at high risk of HPV-related anal disease. People living with human immunodeficiency virus (HIV) infection (PLWH) have an estimated incidence rate of 35 per 100,000 person-years for anal cancer, and a recurrence rate of 50% for high-grade anal intraepithelial neoplasia (HGAIN) within a year [17]. PLWH are at a higher risk of developing HPV-related lesions and cancer, probably due to a lower immunological response and interactions between the viruses [1].

Penile cancer is a rare entity. Globally, it has been estimated that there are 22,000 cases per year [18]; around 40% of penile cancers are due to HPV infection [4]. The most common genotypes detected in penile carcinoma are HPV-16, followed by HPV-18 [19].

At present, there are two HPV vaccines licensed in males. They are composed of virus-like particles and induce immunity against certain HPV types present in the vaccine. The quadrivalent HPV vaccine (4vHPV), first licensed in the EU in 2006, covers HPV-6,11,16,18 and the nine-valent HPV vaccine (9vHPV), licensed in the EU in 2015, which additionally protects against HPV-31,33,45,52,58 [20,21]. In 2014, Austria implemented the 4vHPV vaccine in the standard vaccination schedule for boys and girls, starting a gender-neutral vaccination program. Some countries only followed this approach later, with Uruguay, Germany, and Portugal recommending it in 2019. As routine vaccination of boys is not yet implemented in several countries, exceptionally low coverage of the male population has been reported, with only 4% of men worldwide fully vaccinated as of 2019 [22]. This low HPV vaccination coverage might be associated with the perceived low cost-effectiveness of vaccination. Extending the vaccination to boys and men would cause high financial costs [23]. Lack of HPV vaccine literacy among caregivers, boys, and men [24,25] and the logistical challenges concerning remote populations have also been identified as a barrier to vaccine uptake [26]. Promotion of medical knowledge about the advantages of male HPV vaccination may contribute to improving the management of HPV-related diseases and to increasing vaccine acceptability among parents and boys.

The authors sought to answer the following research question: how effective is the male HPV vaccine, *versus* placebo, in preventing HPV-related disease? This systematic review aimed to assess the available evidence on the impact of male HPV vaccination in preventing HPV-related diseases, which include anogenital warts, recurrent respiratory papillomatosis, anal intraepithelial neoplasia, penile intraepithelial neoplasia, anal and penile cancers, and HNC.

## 2. Materials and Methods

### 2.1. Search Strategy and Study Selection Criteria

This systematic review followed the 2009 Preferred Reporting Items for Systematic Reviews and Meta-analyses (PRISMA) checklist [27]. As eligibility criteria, (1) the study had to investigate the efficacy of vaccination against HPV-related diseases (2) in males of any age, and (3) it had to include a control group. Additionally, it had (4) to report on at least one of the many HPV-related diseases, namely warts/condyloma acuminata in the anogenital area (GW); penile intraepithelial neoplasia (PIN) grade 1, 2, or 3; penile dysplasia; penile cancer; anal intraepithelial neoplasia (AIN) grade 1, 2 or 3; anal dysplasia; anal cancer; head and neck cancer (HNC); and recurrent respiratory papillomatosis (RRP).

The control group had to be males who received placebo or were unvaccinated. For a representative sample of the male population worldwide, we did not exclude men based on sexual orientation and on health status such as HIV status, HPV infection status, or history of HPV-related disease. We excluded (1) studies that evaluated immunogenicity focused on clinical manifestations of disease; (2) studies about infection incidence rates and prevalence, considering that we would not be able to analyze the presence or absence of a relationship between HPV vaccines and HPV-related diseases; (3) studies reporting administration of the vaccine combined with biologically active adjuvants; (4) the use of intralesional HPV vaccine; (5) studies portraying the efficacy of the vaccine by herd protection given by female HPV vaccination; and (6) estimates using statistical models to avoid the influence of confounders on the measurement of the direct efficacy of the vaccination; (7) reviews; (8) systematic reviews; (9) non-randomized controlled trials; (10) cross-sectional studies; (11) case series and (12) case reports; and (13) animal studies (Appendix A).

Our search was focused on high-level scientific evidence of the efficacy of HPV vaccination on clinical manifestations of male disease. We considered studies published from January 2010 to October 2022. No study was excluded based on language.

The searched electronic databases were MEDLINE (PubMed), Web of Science, and Scopus, and were last accessed on 29 October 2022, for all studies. Additionally, ClinicalTrials.gov was searched systematically (last update 22 December 2022) for unpublished or ongoing trials. The queries used are presented in Appendix A, and the terms “HPV”, “human papilloma virus”, “vaccination”, “vaccine”, “immunization”, “efficacy”, “effectiveness”, and “male disease” were the search terms used for the review. Details on the search strategy are reported in Appendix A.

### 2.2. Selection of Studies and Data Extraction 

Screening and a full-text review of the searched studies were performed by two independent investigators (C.R. and C.L.) using the COVIDence platform (COVIDence Systematic Review software, Veritas Health Innovation, Melbourne, Australia), and conflicts were solved by reaching a consensus. 

Two independent reviewers (C.R. and Â.R.F.) used standardized Excel tables to allow the extraction of relevant study data. Conflicts that emerged during the extraction were solved by a third reviewer (C.L.). 

The variables considered important to data extraction included first author and publication year, study design, base study, study site, study period, duration of follow-up, characteristics at enrollment, intervention and comparator, number of participants, inclusion criteria, exclusion criteria, outcomes, results, limitations, sponsorships, and conflicts of interest.

### 2.3. Study Quality Assessment (Risk-of-Bias)

The risk-of-bias assessment was carried out using the following recommended tools: Cochrane risk-of-bias tool for randomized trials (RoB2) [28], and the risk-of-bias in non-randomized studies for interventions (ROBINS-I) [29] for cohort-type studies. Two independent reviewers (Â.R.F. and C.L.) resolved any conflicts over risk-of-bias judgement by reaching a *consensus*. The RoB2 tool assesses five domains of bias in randomized controlled trials: (1) bias arising from the randomization process, (2) bias due to deviations from intended interventions, (3) bias due to missing outcome data, (4) bias in measurement of the outcome, and (5) bias in selection of the reported results. The overall risk-of-bias judgement was categorized either as low, some concerns, or high. 

The ROBINS-I tool comprises seven domains of bias in non-randomized controlled studies: confounding factors, selection of participants for the study, classification of interventions, deviations from intended intervention, missing data, measurement of outcomes, and selection of the reported results. The overall ROBINS-I judgement was categorized as low, moderate, serious, or critical. 

For both tools, the categorization of a domain in a level of risk-of-bias will classify the study at this severity score. 

### 2.4. Glossary of Nomenclature

To homogenize the nomenclatures of the diseases and simplify the reading, we will refer to high-grade anal intraepithelial neoplasia (HGAIN) and high squamous intraepithelial lesion (HSIL) as anal intraepithelial neoplasia (AIN) grade 2/3, and low squamous intraepithelial lesion (LSIL) as AIN grade 1. Similarly, we will refer to high-grade penile intraepithelial neoplasia (HGPIN) and high squamous intraepithelial lesion (HSIL) as penile intraepithelial neoplasia (PIN) grade 2/3, and low squamous intraepithelial lesion (LSIL) as PIN grade 1 [30] (Appendix A). 

## 3. Results

### 3.1. Number of Retrieved Papers

After the query was searched, we initially obtained a total of 632 entries in electronic databases; an additional study was identified in the registry platform ClinicalTrials.gov, resulting in a total of 633 studies. After duplicates (*n* = 182) were eliminated, screening of the title and abstract (*n* = 451) was carried out to select the studies that answered the investigation question. After this screening, 57 studies were eligible for full-text review. From those, 43 studies were excluded according to the following exclusion criteria: assessment of immunogenicity and herd immunity (*n* = 24), wrong study design (*n* = 14), unsuitable interventions (*n* = 3), and finally, evaluation of efficacy only in girls and women (*n* = 2). Consequently, we finally included 13 studies in the study analysis (Figure 1). 

### 3.2. Studies’ General Characteristics (Table 1)

Our systematic review included eight RCTs [31,32,33,34,35,36,37,38], and five prospective cohorts [17,39,40,41,42]. The included cohort studies were sub-studies of RCTs as base studies, except for one [17]. The vaccinated group of participants, in all studies, received a three-dose vaccine regimen [17,31,33,34,35,36,37,38,39,40,41,42], except in one study that considered participants to be vaccinated after receiving one or more doses [32]. Combining all the studies, we obtained data from 14,239 male participants, with 1076 of them being boys aged between 9 and 15 years old at enrollment. Among all the participants, 3502 self-identified as MSM, 832 were PLWH, and 2545 were naïve to all four 4vHPV vaccine genotypes. The age at enrollment ranged, for most studies, from 16 to 26 years; four studies enrolled older men [33,35,38,42]. Seven studies were international [31,32,33,37,39,40,41], while the others occurred in the USA [17,42], Japan [34], Spain [35], The Netherlands [36], and Turkey [38]. 

Twelve studies used the 4vHPV vaccine and a single one used the 9vHPV vaccine [39]. All the included studies assessed the efficacy of the vaccine against disease caused by vaccine types, except the study by Goldstone et al. [28], which failed to demonstrate the efficacy of 4vHPV vaccine against anal disease and EGL caused by ten additional non-vaccine HPV types. 

Two studies evaluated the efficacy of the 4vHPV vaccine on HPV-naïve men [32,35]; one of them enrolled only men that were HPV-naïve to all four genotypes covered by the vaccine [32], while the other had the inclusion criterion of being seronegative for at least two vaccine genotypes [35]. Concerning HPV-related disease history, six studies evaluated vaccine efficacy in men with previous HPV-related diseases [35,36,38], and seven studies included men without history of disease. 

**Table 1 vaccines-11-01083-t001:** Characteristics of the included studies for anal HPV-related diseases and genital HPV-related diseases.

Study	Study Design	Base Study	Study Site	Study Period	Intervention/Comparator	Number of Participants	Characteristics at Enrollment	Inclusion Criteria	Exclusion Criteria	Follow-Up
[37] Giuliano, A. R., 2011 #	RCT NCT00090285	-	Africa, Australia, Europe, and America (71 sites in 18 countries)	2004 to 2008	4vHPV vaccine/placebo	4065 participants (2032 vaccinated and 2033 placebo)	HIV negative	Heterosexual men aged 16 to 23 years old or MSM aged 16 to 26 years old, all with 1 to 5 sexual partners in their lifetime	Clinically detectable anogenital warts or genital lesions at screening or who had a history of such findings	2.9 years
[31] Palefsky, J. M., 2011 ∆	RCT	Giuliano A.R et al. [38] NCT00090285	Australia, America, and Europe (7 countries)	2004 to 2008	4vHPV vaccine/placebo	598 participants (299 vaccinated and 299 placebo)	MSM+HIV negative; 16 to 26 years old	MSM aged 16 to 26 years old, with five or less sexual partners and anal intercourse/oral sex with other male in the last year	Medical history anogenital warts, genital lesions suggesting other STI, or intra-anal lesion related to AIN or condyloma. HIV-positive men at enrollment	3 years
[17] Swedish, K. A., 2012 ∆	Prospective Cohort	-	USA, New York City (1 site)	2007 to 2010	4vHPV vaccination/unvaccination	202 participants (88 vaccinated and 114 unvaccinated)	MSM+HIV negative; ≥18 years old	MSM aged 18 years old or older, HIV-negative, and with clinical history of biopsy-proven and treated HGAIN **	Patients with HGAIN ** at the beginning and who had not received all 3 doses of 4vHPV vaccine	3 years
[32] Goldstone, S. E., 2013 ∆ #	RCT	Giuliano A.R. et al. [38] NCT00090285	Africa, Australia, Europe, America (71 sites in 18 countries)	2004 to 2009	4vHPV vaccine/placebo	4055 participants (2025 vaccinated and 2033 placebo)	Heterosexual males aged 16 to 24 years old MSM aged 16 to 27 years with a maximum of 6 lifetime sexual partners	For the HPV-naïve population, patients who were PCR-negative to all the 14 tested HPV types and seronegative to HPV-6/11/16/18. MSM with a normal anal cytology test. For ITT population, participants who received at least one dose of vaccine or placebo and returned for at least one follow-up visit	Immunodeficiency or HIV infection upon enrollment. Clinical history of genital lesions related to HPV infection or other sexually transmitted infection. Presence of lesions of unknown etiology	3 years
[42] Swedish, K. A., 2014 ∆	Prospective Cohort	Swedish K. A. et al. Clin Infect Dis 2012; 54:891–898.	USA, New York City (1 site)	2007 to 2010	4vHPV vaccine/unvaccinated	313 participants (116 vaccinated and 197 unvaccinated)	MSM+HIV negative; ≥26 years	MSM+HIV- aged up to 26 years old. No clinical history of anal condyloma, or recurrence free for at least a year	Patients who did not receive all 3 doses of vaccine and those vaccinated at other sites. Patients with anal or penile condyloma at the start of the study	4 years
[40] Ferris, D., 2014 ∆ #	Prospective Cohort	Reisinger K. S. et al. Pediatr Infect Dis J 2007; 26(3): 201-209 NCT00092547	Europe, America, Asia, Africa (72 sites in 17 countries)	2005 to 2013	4vHPV vaccine	775 participants (555 EVG and 220 CVG)	9 to 15 years old	EVG: boys who received 4vHPV vaccine between the ages of 9 and 15 years old during the base study CVG: boys who received placebo in the base study and later received the 4vHPV vaccine.	Allergy to any vaccine component, thrombocytopenia, immunosuppression/previous immunosuppressive therapy, or previous receipt of an HPV vaccine	8 years
[38] Coskuner, E. R., 2014 #	RCT	-	Turkey (1 site)	2009 to 2013	4vHPV vaccine/placebo	171 participants (91 vaccinated and 80 placebo)	Circumcised men with only female partners	Men with new onset of genital warts living in the same area for at least 1 year	Clinical history of treatment of pre-existing warts, chronic treatment for medical disorders or immunosuppression (including HIV)	4 years
[33] Wilkin, T. J., 2018 ∆	RCT NCT01461096	-	USA and Brazil (24 sites in 2 countries)	2012 to 2015	4vHPV vaccine/placebo	577 participants (288 vaccinated and 287 unvaccinated)	HIV+; ≥27 years old	Men aged 27 years old or older, HIV+, who had had anal intercourse/oral sex with another male in the last year	History of HPV-related cancer, anal HSIL or condyloma treatment in the last 6 months. Prior HPV vaccination or allergy to vaccine components. Anticoagulant use, active drug, or alcohol use. Bleeding diatheses, systemic anti-neoplastic or immunomodulatory treatment	3 to 4 years
[34] Mikamo, H., 2019 #	RCT NCT01862874	-	Japan (24 sites)	2013 to 2017	4vHPV vaccine/placebo	1124 participants (562 vaccinated and 562 placebo)	Heterosexual or MSM aged 16 to 26 years old Participants refrained from sexual activity for 2 days before scheduled visits that included sample collection	Heterosexual men aged 16-26 years old, with 1 to 5 exclusively female partners; MSM aged 16-26 years who had had anal intercourse/oral sex with a male partner within the past year, and with 1 to 5 lifetime male/female partners.	History of genital warts or clinically present external genital warts at the beginning of the study	3 years
[39] Olsson, S. E., 2020 ∆ #	Prospective cohort	P. Van Damme et al. Pediatrics 2015; 136: e28–e39 NCT00943722	Europe, America, Asia, Africa (39 sites in 13 countries)	2013 to 2018	9vHPV vaccine	301 participants	9 to 15 years old	Boys aged 9 to 15 years old who received all 3 vaccine doses during the base study	Clinical history of HPV infection. Allergy to any vaccine component, thrombocytopenia, immunosuppression/previous immunosuppressive therapy, or previous receipt of an HPV vaccine. Current enrollment in any other clinical study	10 years
[35] Hidalgo-Tenorio, C., 2021 ∆ #	RCT	Hidalgo-Tenorio C. et al. AIDS Res Ther 2017; 14: 34 ISRCTN14732216	Spain (1 site)	2011 to 2017	4vHPV vaccine/placebo	129 participants (66 4vHPV vaccinated and 63 placebo)	MSM+HIV+; >26 years old	MSM aged > 26 years old, HIV-positive, and not infected with all 4vHPV vaccine types at the same time, or not infected with genotypes 16 and 18 at the same time. Normal high resolution anoscopy, anal biopsy with condyloma alone, anal LSIL ** alone	MSM+HIV+ patients with simultaneous anal infection with the four genotypes addressed by the vaccine, and who at least had HPV genotypes 16 and 18. Active opportunist infection at enrollment. Patients who had HSIL **, or ASCC or had received treatment for these lesions. History of allergy to aluminum and/or yeast extract excipient	4 years
[36] Gosens, K. C. M., 2021 ∆	RCT NCT02087384	-	Amsterdam, The Netherlands (3 sites)	2014 to 2018	4vHPV vaccine/placebo	126 participants (64 vaccinated and 62 placebo)	MSM+HIV+; ≥18 years old	HIV+MSM aged 18 years old or older who had a CD4+ cell count greater than 350 cells/µL, had biopsy-proven intra-anal HGAIN ** (successfully treated), and had lesions still in remission (maximum LGAIN **) at the time of enrollment	Previous HPV vaccination, allergy to any 4vHPV vaccine constituents, and other comorbidities. Medical history of anal carcinoma, current peri-anal HGAIN ** or peri-anal AIN2 or 3 at the time of enrollment. Immunosuppressive medication or immunodeficiency other than HIV, or a life expectancy inferior to a year	1.5 years
[41] Goldstone, S. E., 2022 ∆ #	Prospective Cohort	Giuliano A.R. et al. [38] NCT00090285	Australia, America, Europe, Asia, Africa (46 sites in 16 countries)	2010 to 2017	4vHPV vaccine	1803 participants (936 EVG and 867 CVG)	16 to 26 years old	EVG: Heterosexual men aged 16 to 23 years old and homosexual men aged 16 to 26 years old who received 1 or 2 doses or more of the 4vHPV vaccine at the base study. CVG: Heterosexual men aged 16 to 23 years old and homosexual men aged 16 to 26 years old who received 1 or 2 doses or more of the 4vHPV vaccine, and were placebos in the base study	Enrollment in a study that involved/interfere with the collection of anogenital samples. History of anogenital warts or genital lesions	EVG: 9.5 years CVG: 4.7 years

Abbreviations: 4vHPV vaccine = Quadrivalent HPV vaccine; AIN = Anal intraepithelial neoplasia; ASCC = Anal squamous cell carcinoma; CVG = Catch-up vaccination group; EVG = Early vaccination group; HGAIN = High-grade anal intraepithelial neoplasia; HIV = Human immunodeficiency virus; LGAIN = Low-grade anal intraepithelial neoplasia; LSIL = Low-grade intraepithelial lesion; MSM = Men who have sex with men; PCR = Polymerase chain reaction; RCT = Randomized controlled trial; STI = Sexually transmitted infection. ** Note: HGAIN and HSIL corresponds to AIN grade 2/3; LSIL corresponds to AIN grade 1; Penile HGAIN and HSIL corresponds to PIN grade 2/3; penile LSIL corresponds to PIN grade 1 [30]. ∆ Anal HPV-related diseases; # Genital HPV-related diseases.

Of our predefined outcomes, ten studies assessed the occurrence of condyloma acuminata [31,32,34,35,37,38,39,40,41,42]; six provided data on PIN and penile cancer [32,34,37,39,40,41]; and ten evaluated AIN and anal cancer [17,31,32,33,35,36,39,40,41,42]. 

We did not find studies that had assessed other disease outcomes, such as RRP or HNC, related to HPV in the male population. 

Conflicts of interest were disclosed in ten studies, while the remaining three had none [36,39,41]. A pharmaceutical company sponsored seven studies [31,32,34,37,39,40,41] and donated vaccines to one of them [36] (Table 2 and Table 3).

### 3.3. Outcomes Reported in the Included Studies of HPV-Related Anal Disease 

Ten of the included studies focus on anal male disease; data from these studies are summarized in Table 2.

Seven studies presented efficacy against anal HPV-related disease [17,31,32,39,40,41,42]. Participants’ age ranged from 16 to 26 years old, except in one study that had enrolled men older than 26 years [42]. Efficacy against anal condyloma acuminata was assessed by a cohort [42] that showed a significantly lower incidence rate in vaccinated participants, with an HR = 0.45. Two RCTs [31,32] and one cohort [41] assessed efficacy against AIN1. In the RCTs, efficacy ranged from 91.1% (including condyloma acuminata) [31] to 93.1% (not including condyloma acuminata) [32]. In the cohort study, a sub-study based on a previous RCT, the early and the catch-up vaccination groups (EVG and CVG, respectively) both had a lower incidence rate of anal disease when compared to the placebo group of the base study [41]. Efficacy against AIN2|3 was assessed using two RCTs; one RCT reported an efficacy of 91.7% against AIN2|3 [31], and the other one an efficacy of 89.6% or worse against AIN2|3 in the HPV-naïve population, and 50.3% in the intention-to-treat (ITT) population [32]. A cohort reported that vaccinated participants revealed a lower incidence rate and lower risk of recurrence of AIN2|3 at 1 year (HR = 0.42), and similar risks at years 2 and 3 (HR = 0.50, and HR = 0.52, respectively) [17]. Regarding anal cancer outcomes, only one cohort [41] reported a protective effect in vaccinated men, both in EVG and CVG groups. Of the aforementioned studies, four studies included male participants without history of HPV disease, and all of them were HIV-negative at enrollment [31,32,39,40]; two of them assessed efficacy on adolescents [39,40], and one of them included only MSM participants [31]. The other studied both heterosexual men and MSM [32]. Three studies enrolled men with history of disease, all HIV-negative men; two studies included only MSM [17,42], while a third one included both MSM and heterosexual men [41].

Three studies showed no efficacy; participants in these studies were ≥26 years old [33,35,36]. Three RCTs [33,35,36] assessed efficacy for AIN2|3, and none of them showed significant differences between vaccinated and unvaccinated male participants. Considering anal cancer outcomes, as assessed by two RCTs, none found significant differences between vaccinated men and control groups. Two studies [35,36] included MSM living with HIV infection (PLWHMSM) with history of HPV disease. 

**Table 2 vaccines-11-01083-t002:** Overview of anal HPV-related outcomes reported in the studies.

Study	Outcomes	Results	Limitations	Sponsorship/Conflicts of Interest	Risk-of-Bias
[31] Palefsky, J. M., 2011	Efficacy of 4vHPV vaccine on the prevention of any grade of AIN or anal cancer related to infection with HPV-6/11/16/18	The PP efficacy of the 4vHPV vaccine against AIN1 (including condyloma) was 91.1% and for AIN 2 or 3 of 91.7%. The ITT population efficacy was 50.3%, both for AIN1/2/3	Narrow range of ages. Short follow-up time. Participants had limited sexual activity. Findings may not be generalizable to boys and men in the general population of similar ages	Sponsorship: Merck CI: Disclosed	Low
[17] Swedish, K. A., 2012	Efficacy of 4vHPV vaccine in the prevention of recurrent HGAIN among MSM+HIV-	Hazard analysis revealed an increased risk of recurrent HGAIN **, but 4vHPV vaccination was associated with a decreased risk. The vaccinated patients have an incidence rate of 10.2 per 100 person years for recurrent HGAIN ** and unvaccinated patients have rates of 15.7 per 100 person-year. Comparing the risk of recurrent HGAIN ** on 4vHPV vaccinated individuals, the multivariable model for different years presents lower risk at 1 years (HR = 0.42) and similar risk at 2 and 3 years (HR = 0.50 and HR = 0.52, respectively). The Kaplan–Meier’s survival analysis indicates no recurrence of HGAIN ** in vaccinated individuals	The study depended on pre-existing medical records lacking race/ethnicity and oncogenic HPV infection status, and it was difficult to determine a “study entry” for unvaccinated participants. The sample comprises a mostly white population of non-smoking MSM with private insurance, so findings may not be generalizable to other populations	CI: Disclosed	Moderate
[32] Goldstone, S. E., 2013	Efficacy of 4vHPV vaccine against AIN caused by HPV-6/11/16/18 and by 10 additional non-vaccine HPV types as well as efficacy regardless of whether HPV was detected in an AIN lesion.	A protective effect of vaccination on HPV 6/11/16/18-related AIN and anal cancer was identified. In the ITT, efficacy was 50.3% and, in the HPV,-naïve group, and was 89.6% regarding 6/11/16/18-related anal disease. Efficacy against AIN 1 in the naïve group was statistically significant (93.1%). Efficacy against anal disease endpoints caused by the ten non-vaccine types was not demonstrated	Young participants with limited number of sexual partners. Small sample of MSM; Short medium follow-up < 3 years. Low evidence due to low rate of events such as anal cancer	Sponsorship: Merck & Co, Inc. CI: Disclosed	Low
[40] Ferris, D., 2014	Efficacy of 4vHPV vaccination against HPV-6/11/16/18-related disease	In both groups, there were no cases of HPV-6/11/16/18-related disease, including perianal cancer	Small sample size and observed attritions	Sponsorship: Merck Sharp & Dome CI: None	Low
[42] Swedish, K. A., 2014	Efficacy of 4vHPV vaccine in the prevention of recurrent anal condyloma among MSM+HIV-	Anal condyloma cases appeared in 8.6% of 4vHPV vaccinated participants, and 18.8% unvaccinated participants. Vaccination was associated with decreased risk of anal condyloma (HR = 0.45)	Groups have considerable differences upon enrollment. Sample size of unvaccinated patients was larger, and they were followed over a longer period	CI: Disclosed	Moderate
[33] Wilkin, T. J., 2018	Efficacy of 4vHPV vaccine on HSIL and anal cytological outcomes	Anal HSIL ** is not influenced by the 4vHPV vaccine (0% efficacy), and anal cytology was not statistically different between groups	Early suspension by DSMB due to protocol-defined futility rules. No access/approval for the nine-valent HPV vaccine at the beginning. No review of cytology or histology outcomes	CI: Disclosed	Moderate
[39] Olsson, S. E., 2020	Incidence of the composite endpoint of HPV-6/11/16/18/31/33/45/52/58-related perianal cancer ≥ 6 months duration	According to the PP analysis, there were three HPV-related diseases, including perianal cancer	Control group from the base study	Sponsorship: Merck Sharp & Dome CI: Disclosed	Moderate
[35] Hidalgo-Tenorio, C., 2021	Effectiveness of the 4vHPV vaccine to prevent anal ≥ HSILs by 4vHPV vaccine genotypes in MSM+HIV+	Vaccinated and unvaccinated individuals have the same risk of development of ≥HSILs **; however, the 4vHPV vaccine showed a protective effect against HPV6 during the first year of follow-up. The factor associated with the risk of anal ≥ HSIL ** was receipt of the last dose of the vaccine less than 6 months earlier, in comparison to those vaccinated for a longer period	Strict eligibility criteria	CI: None	Low
[36] Gosens, K. C. M., 2021	1. Cumulative recurrence of intra/peri-anal HGAIN (biopsy-proven) at 12 months after last vaccination 2. Recurrence of intra/perianal HGAIN at time of last vaccination and at 6 months after last vaccination, cumulative occurrence of LGAIN, or anogenital condyloma, causative HPV genotype in recurrent HGAIN lesions, and safety of the 4vHPV vaccine	Cumulative HGAIN ** recurrence is similar between 4vHPV and placebo individuals. The recurrence of HGAIN ** has an incidence rate of 66.3 per 100 person years for 4vHPV vaccinated individuals and 56.5 for placebo. From the 78 recurrences of HGAIN **, only one was peri-anal, and no progression to anal cancer was identified during follow-up	Short follow-up time. Microscopical lesions were undetected/misdiagnosed (but equally distributed using randomization)	Sponsorship: Merck Sharp & Dome CI: Disclosed	Moderate
[41] Goldstone, S. E., 2022	Incidence per 10,000 persons years of AIN or anal cancer related to HPV-6/11/16/18 in MSM	For the EVG, there were no new cases of HGAIN ** related to HPV-6, 11, 16, and 18 during the present study. For CVG, the incidence per 10,000 person years was lower during the long-term follow-up period than during the base study period for AIN and anal cancer related to HPV-6/11/16/18. In addition, the incidence of anal disease in both the EVG and CVG during the long-term follow-up period was significantly lower than in placebo recipients during the base study	High loss of participants	Sponsorship: Merck Sharp & Dome CI: Disclosed	Low

Abbreviations: 4vHPV vaccine = Quadrivalent HPV vaccine; AIN = Anal intraepithelial neoplasia; ASCC = Anal squamous cell carcinoma; CVG = Catch-up vaccination group; EVG = Early vaccination group; HGAIN = High-grade anal intraepithelial neoplasia; HIV = Human immunodeficiency virus; LGAIN = Low-grade anal intraepithelial neoplasia; LSIL = Low-grade intraepithelial lesion; MSM = Men who have sex with men; PCR = Polymerase chain reaction; RCT = Randomized controlled trial; STI = Sexually transmitted infection. ** Note: HGAIN and HSIL corresponds to AIN grade 2/3; LSIL corresponds to AIN grade 1 [30].

### 3.4. Outcomes Reported in the Included Studies of HPV-Related Genital Disease

Eight studies analyzed male genital diseases attributable to HPV; data are summarized in Table 3. We found five studies that reported efficacy [32,37,39,40,41]; at enrollment, participants’ ages ranged from 16 to 27 years old. Efficacy against genital condyloma acuminata and PIN1|2|3 was assessed by five studies, two RCTs [32,37], and three cohorts [39,40,41]. One RCT [33] reported an efficacy of 89.9% against genital condyloma for the HPV-naïve population, and 66.7% for the ITT population, while the other RCT [37] reported an efficacy of 67.2% in the ITT population. Concerning the PIN outcome, both RCTs found no significant differences between vaccinated and placebo groups [32,37]. All three cohort studies evaluated the same outcomes, genital condyloma and PIN, and reported a decrease in the incidence rate in vaccinated participants [39,40,41]. Regarding penile cancer, four studies assessed this outcome: an RCT [37] and three cohorts studies [39,40,41]. However, efficacy could not be assessed, as there were no cases diagnosed. The cohort studies reported low incidence rates. Four of the studies mentioned above [32,37,39,40] included HIV-negative men without a history of HPV disease; two studies within these enrolled adolescents [40,41]. Heterosexual men and MSM were eligible in two studies [32,41].

Three RCTs showed no efficacy of the HPV vaccine against genital condyloma [34,35,38]. Participants were 26 years or older at enrollment [35,38], except in one study [34] (age ranged from 16 to 26 years). One of the RCTs assessed PIN1 including condyloma and reported not having enough statistical power to evaluate both outcomes [34]. None of these studies assessed PIN2|3 or penile cancer. Two of the studies had participants with a history of HPV disease [35,38]; one included PLWH and MSM participants [35], while the other enrolled only HIV-negative heterosexual men [38]. One study included both heterosexuals and MSM participants without a history of HPV disease and with unknown HIV status [34].

### 3.5. Study Quality Assessment (Risk-of-Bias)

For RCT studies, topics concerning randomization after entry, blindness of treatment, and loss to follow-up are of major importance for risk-of-bias assessment. In cohort studies in which some factors are not controlled, evaluation of potential confounders, such as biases in participants’ selection and measurement of outcomes, is of paramount significance.

Regarding risk of bias in this systematic review, the studies were categorized as having low, moderate, or high risk of bias (Table 2 and Table 3). Five studies on anal HPV-related diseases were classified as having moderate risk of bias, and the remaining five as having low risk of bias. Concerning genital HPV-related diseases, three were categorized as having moderate risk of bias, and five as having low risk of bias. 

Risk of bias is associated with different methodologies used between patients within groups, their blinding to treatment, suspension of the studies, the reliability of the diagnostic techniques applied, and the different times from entry to follow-up.

**Table 3 vaccines-11-01083-t003:** Overview of genital HPV-related outcomes reported in the studies.

Study	Outcomes	Results	Limitations	Sponsorship/Conflicts of Interest	Risk-of-Bias
[37] Giuliano, A. R., 2011	Reduction in the incidence (as compared with placebo) of EGL associated with HPV present in 4vHPV vaccine or to any HPV type.	The ITT efficacy to reduce recurrent EGL was 60.2% overall; in particular, 4vHPV vaccine has 67.2% of efficacy in preventing condyloma. The PP analysis showed an overall 83.8% efficacy and 89.4% 4vHPV vaccine efficacy for condyloma. Penile cancer was not evaluated due to lack of cases in the sample, and a low number of participants presented PIN	Short follow-up time and narrow age range of the subjects	Sponsorship: Merck & Co, Inc. CI: Disclosed	Moderate
[32] Goldstone, S. E., 2013	Efficacy of 4vHPV vaccine against EGL caused by HPV-6/11/16/18 and by 10 additional non-vaccine HPV types.	4vHPV vaccine had a protective effect against EGL. Although significant efficacy against non-vaccine HPV types was not seen. Efficacy was 9.3% lower at external genital sites when compared with efficacy against HPV 6/11/16/18	Young participants with limited number of sexual partners. Small sample of MSM. Short medium follow-up < 3 years	Sponsorship: Merck & Co, Inc. CI: Disclosed	Low
[38] Coskuner, E. R., 2014	Efficacy of 4vHPV vaccination on the recurrence of genital warts	There was no difference in recurrence between vaccinated and unvaccinated group, except for the marital status and number of lesions	Small sample size and only from a center. Absence of an initial assessment of vaccine-type antibodies or DNA. No data about previous HPV exposure	CI: None	Moderate
[40] Ferris, D., 2014	Efficacy of 4vHPV vaccination against HPV-6/11/16/18-related disease	In both groups, there were no cases of HPV-6/11/16/18-related disease: genital condyloma, PIN, or penile cancer	Small sample size and observed attritions	Sponsorship: Merck Sharp & Dome CI: None	Low
[34] Mikamo, H., 2019	Efficacy of 4vHPV vaccination on the reduction in the combined incidence of HPV-6/11/16/18-related condyloma acuminata, PIN, and penile, perianal, or perineal cancer.	Vaccination had 83.4% efficacy at interim analysis, and at the final analysis, 86.5% efficacy against combined incidence of HPV-6/11/16/18-related persistent infection and EGL. Only at the final analysis were cases of EGL detected in the placebo group (condyloma and PIN1)	High loss of participants and short follow-up time. This study does not have sufficient power to evaluate the disease endpoint (condyloma, PIN)	Sponsorship: Merck Sharp & Dome CI: Disclosed	Low
[39] Olsson, S. E., 2020	Incidence of the composite endpoint of HPV-6/11/16/18/31/33/45/52/58-related PIN, genital warts and penile cancer of ≥6 months duration	According to the PP analysis, there were three HPV-related disease, including perianal cancer.	Control group from the base study	Sponsorship: Merck Sharp & Dome CI: Disclosed	Moderate
[35] Hidalgo-Tenorio, C., 2021	Effectiveness of the 4vHPV vaccine to prevent anal ≥ HSILs, and EAGLs in MSM+HIV+	Vaccinated and unvaccinated individuals have the same risk of development of ≥HSILs or EAGL; however, the 4vHPV vaccine showed a protective effect against HPV6 during the first year of follow-up	Strict eligibility criteria	CI: None	Low
[41] Goldstone, S. E., 2022	Incidence per 10,000 persons years of external genital warts related to HPV-6/11 and external genital lesions related to HPV-6/11/16/18 in all participants	For the EVG, when compared to placebo group of the base study, results of incidence were null for external genital warts related to HPV-6/11 and external genital lesions related to HPV-6/11/16/18. For CVG, the incidence per 10,000 person years was lower during the long-term follow-up period than during the base study period for external genital warts related to HPV-6/11, and external genital lesions related to HPV-6/11/16/18. Comparing EVG and CVG in the present study, the incidence of the external genital warts related to HPV-6/11 and external genital lesions related to HPV-6/11/16/18 was similar to the incidence in the EVG	High loss of participants	Sponsorship: Merck Sharp & Dome CI: Disclosed	Low

Abbreviations: 4vHPV vaccine = Quadrivalent HPV vaccine; CI = Conflict of Interest; CVG = Catch-up vaccination group; EAGL = external ano-genital lesions; EGL = external genital lesions; EVG = Early vaccination group; MSM = Men who have sex with men; PIN = penile/perianal/perineal intraepithelial neoplasia; PP = Per-protocol. Note: Penile HGAIN and HSIL corresponds to PIN grade 2/3; penile LSIL corresponds to PIN grade 1 [30].

## 4. Discussion

We conducted a systematic review that assessed the efficacy of HPV vaccination in male disease. Overall, we found that HPV vaccine in males was effective, although the characteristics of the study participants and the outcomes assessed influenced the impact of vaccination upon HPV-related anogenital disease. 

The participants’ age and history of HPV-related disease influenced vaccination efficacy. We observed higher efficacy in men who were naïve to the respective HPV types evaluated in the studies. Interestingly, the seroprevalence of any HPV vaccine type increased with age [43,44]. Thus, the likelihood of being HPV-naïve is higher in childhood/early adolescence before exposure to HPV through sexual activity. Accordingly, the studies that assessed vaccine efficacy in males older than 26 years and with previous history of HPV-related disease found no efficacy against genital disease [35,38]. The study by Goldstone S. et al. [41] reported a similar incidence of EGL in the EVG (young men aged 16 to 26) and in the CVG (men older than 26 years). However, the participants of the CVG were young men belonging to a placebo group in a previous RCT [37], and history of HPV-related genital lesions was one of the exclusion criteria. On the other hand, for anal disease, two studies assessed men with previous history of disease and showed efficacy [17,42], while two others reported no efficacy in these participants [35,36]. In fact, studies that aimed to assess the efficacy of 4vHPV vaccine in the prevention of recurrent AIN2|3 [17,36] reported different results. Swedish et al. [17] found a decreased risk of AIN2|3 recurrence in vaccinated individuals, while Gosens et al. [36] found no efficacy of the 4vHPV vaccine in the prevention of AIN2|3 recurrence. Notably, this last study enrolled HIV-positive men and the follow-up was very short (1.5 years) which might affect these findings. Wilkin et al. [33] evaluated the efficacy of the 4vHPV vaccine against anal HSIL in HIV-positive men aged 27 years or older; about a third of them had a history of anal disease, and no efficacy of 4vHPV vaccine was found. Therefore, these studies do not support the ability of the HPV vaccine in HIV-positive men aged 27 years or older to prevent or improve anal disease outcomes. HPV vaccination provides protection against persistent HPV infections and against the development of AIN2|3 in individuals without prior vaccine-type HPV infection [45]. However, catch-up vaccination is also recommended for MSM and HIV-infected individuals up to 26 years of age [46].

Regarding data in males aged between 16 and 26 years, (CVG) Ferris et al. [40] and Goldstone et al. [41] reported HPV vaccine efficacy against genital condyloma, PIN and AIN2|3, although the results were inferior to those of the younger group (EVG). Both studies were of high quality (and low risk of bias), thereby allowing generalizations to other populations and supporting the current recommendation for HPV vaccination of men up to age 26 years [47].

Regarding vaccine efficacy in the prevention of HPV-related anogenital diseases, the included studies found that the highest vaccine efficacy was observed for AIN 1|2|3 [17,31,32,40,41,42], followed by genital condyloma [32,37,39,40,41] and anal condyloma [31,40,41,42]. The studies that assessed these outcomes have from high to fair quality (i.e., low to moderate risk of bias). In contrast, a single study [40] assessed vaccine efficacy against PIN. The authors did not find any protective effect of the HPV vaccine against HPV-6, 11, 16 and 18 related to PIN. However, this study has critical limitations: the small size of the study population, and collection of genital specimens in several centers without standardization of the procedure. In fact, male genital sampling has not been standardized as it has in women [48]. It should be noted that the retrieved studies couldn’t assess vaccine efficacy/effectiveness against penile or anal cancer [37,39,40,41] due to absence of cases present during the studies. The follow-up ranged from 2.9 years [37] to 10 years [39].

PLWH and MSM have a greater risk of developing disease associated with HPV. HIV infection can reduce vaccine immunogenicity and effectiveness due to low immune response in these patients [1]. MSM may be at greater risk of disease, as they do not benefit from the herd immunity provided by female vaccination against HPV. 

Vaccinating women can protect heterosexual men, but this herd protection does not extend to MSM and bisexual men [49]. Not considering this group for vaccination can contribute to an increase HPV-related diseases, especially anogenital cancer [50,51]. Barriers to vaccine uptake in these groups do arise from costs, and from incorrect association of HPV vaccine with cervical cancer alone [52]. Additionally, a lack of physicians’ recommendations, at times due to non-disclosure of patients’ sexual orientation [53], contributes to low HPV vaccine coverage.

Our systematic review included studies that enrolled PLWH, although none of them reported on the difference in HPV vaccine efficacy between PLWH and HIV-negative people. The study by Bergman et al. and two systematic reviews by Zhan et al. found that HPV vaccination is effective and safe in PLWH [54,55]. However, they highlighted that vaccine efficacy in the prevention of HPV-related neoplasia remains unknown. In our review, the studies that showed efficacy against anal and genital HPV-related disease enrolled HIV-negative men. Three studies [35,37,38] included PLWH, and reported that vaccinated and unvaccinated men had the same risk of AIN2|3. Additionally, Gosens et al. [36] reported similar cumulative AIN2|3 recurrence between vaccinated men and the placebo group. 

Our review supports previous findings, namely regarding higher HPV vaccine efficacy in boys before sexual debut [56]. Moreover, HPV vaccination in males is effective against AIN1|2|3 [57] and anogenital warts [57,58]. As shown in our systematic review, the body of evidence is much weaker regarding penile precancerous lesions in males.

One of the strengths of this systematic review results from the number of participants analyzed, which reached 14,239 boys and men. Another strength is the fact that we were able to include a diverse study population (MSM as well as heterosexual men, HIV-negative men and PLWH, HPV-naïve males and men with history of HPV-related disease) from several countries worldwide, meaning we could observe real-life scenarios and allowing the generalization of the findings. Our review was conducted based on internationally accepted methods, and the tools used for evaluation of the studies’ risk-of-bias are part of the recommended ones. Concerning the quality assessment using RoB2 and ROBINS-I tools, the two independent reviewers reached a substantial agreement [59,60]

During the selection process, we came across to some publications assessing the impact of the vaccine on RRP and HNC, but they failed to meet the defined inclusion criteria. 

The limitations of our review mainly arise from the limitations of the included studies. Overall, the studies included males of a narrow range of age, mostly young males with somewhat limited sexual activity, thereby challenging the generalization of the results to other populations [17,31,32,37,42]. On the other hand, several studies analyzed small population samples from the outset, while others reported a smaller population size at the end of their studies, as dictated by discontinuation [34,38,40,41]. Most of the studies had short follow-up times, and no meaningful conclusions could be carried out for HPV-related premalignant and malignant lesions. We also noticed that for genital outcomes, most of the participants were heterosexual men, possibly leading to an underestimation of the vaccine efficacy itself, since these participants could benefit from herd immunity. Finally, the vaccine efficacy in the existing studies was expressed using absolute risk reductions, relative risk reductions, and a hazard ratio, and sometimes the incidence rate is the sole result. It became difficult and sometimes impossible to compare results and analyze the magnitude of the variations in efficacy among studies, which could have helped to assess their true significance. Additional limitations of our systematic review may also arise from the restriction of the outcomes to the clinical manifestations only. Our query could be made more inclusive by including synonyms of efficacy. However, reviewing all the references in the literature, we have not found any additional eligible study meeting the inclusion and exclusion criteria. A further limitation concerns the strict inclusion and exclusion criteria, but as stated at the beginning of our study, we aimed for high-quality evidence of the efficacy of HPV vaccination on male disease, not including immunogenicity studies. 

## 5. Conclusions

HPV-related diseases in the male population represent an emerging problem, resulting in morbidity and mortality and associated with huge health costs due to recurrence and cancer risk. Finding a way to prevent them is a crucial public health strategy. Increasing vaccination coverage, regardless of sexual orientation or gender identity, and monitoring the impact of vaccination should become public health priorities.

This systematic review supports the recommendation of early vaccination of boys before the onset of sexual activity. There is clear evidence of the HPV vaccine’s efficacy in men up to the age of 26 years old, with or without history of HPV-related disease. HPV vaccination protects against anal and genital warts and AIN1|2|3.

It is worth considering the impact of the HPV vaccine on HPV-related penile cancer and PIN in future studies. Moreover, in the future, it will be important to carry out RCTs that include more MSM and PLWH, and to evaluate the efficacy of vaccination on genital disease. An unanswered question in this systematic review involves the vaccine’s efficacy against RRP and HNC; this is due to the lack of RCT studies addressing this topic.

## Figures and Tables

**Figure 1 vaccines-11-01083-f001:**
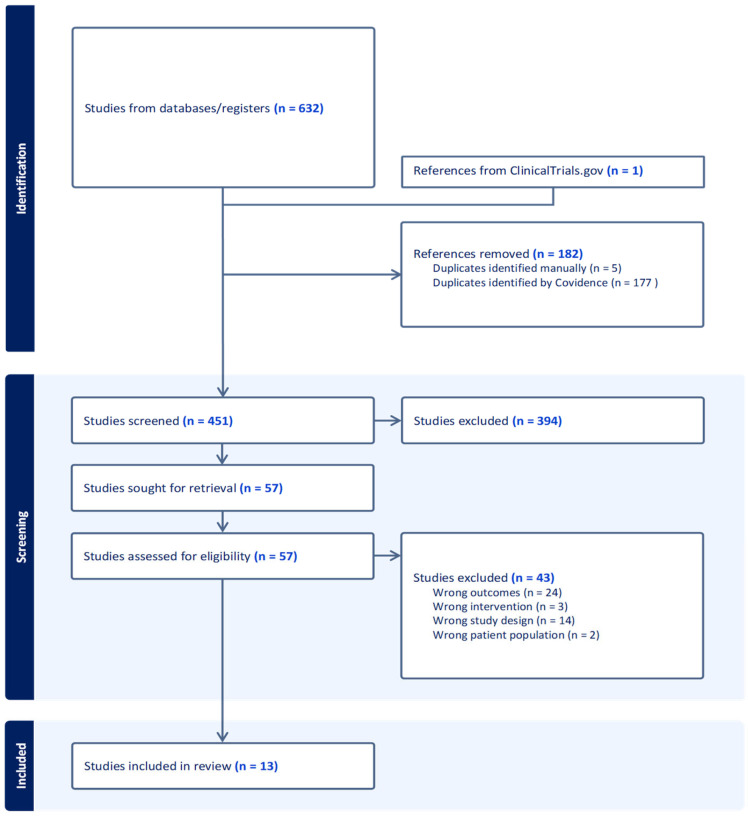
Study selection process flowchart.

## Data Availability

Not applicable.

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
