# Peer review of "Impact of Human Papillomavirus Vaccination on Male Disease: A Systematic Review"

_vaccines, 2023, doi:10.3390/vaccines11061083_

Round 1
Reviewer 1 Report
Overall, this research article presents a systematic review of the impact of the HPV vaccine in male disease, focusing on outcomes related to anal and genital HPV-related diseases. The authors report a detailed methodology for their literature search and selection process, including the number of retrieved papers, the number of studies assessed for eligibility, and the number of included studies. However, there are some areas where the article could be improved, as described below.
Introduction:
The introduction provides a comprehensive overview of the prevalence of HPV-related diseases in males worldwide, highlighting the specific genotypes involved in different diseases. However, the introduction could benefit from a clearer and more structured presentation of the research question and objectives. The research question and objectives are briefly mentioned in the last sentence of the introduction, but a clearer statement of the research question at the outset would have better contextualized the subsequent discussion. Additionally, while the introduction provides information on the prevalence of HPV vaccination in males worldwide, it could benefit from further discussion on the barriers to vaccine uptake in males and the potential impact of increasing vaccine coverage in reducing the burden of HPV-related diseases. Overall, the introduction provides a good overview of the topic but could benefit from greater clarity and structure.
Materials and methods:
Overall, the materials and methods section of this research article appears to be well-structured and clearly presented. However, there are several areas where the methodology could be improved or clarified, which I will discuss in detail below.
Search Strategy and Study Selection Criteria:
The authors have stated that they followed the PRISMA checklist, which is a standard guideline for systematic reviews. However, they have not provided enough detail on the search strategy and study selection criteria to enable the reader to fully evaluate the quality of the review. For example, the authors state that they searched three electronic databases and ClinicalTrials.gov, but they do not specify the search terms used or the search date range for each database. It would also be helpful to know how many studies were identified from each database, how many were screened, and how many were included in the final analysis. Additionally, the authors should provide a flow diagram illustrating the study selection process. Furthermore, the authors state that they included studies that investigated the efficacy of vaccination against HPV-related diseases in males of any age and that had a control group of males who received placebo or were unvaccinated. However, it is unclear why the authors did not include studies that evaluated immunogenicity and infection incidence rates and prevalence, as these are important outcomes for assessing vaccine efficacy. The authors should provide a justification for this exclusion criterion.
Study Quality Assessment (Risk-of-bias):
The authors have stated that they used the Cochrane risk-of-bias tool for randomized trials (RoB2) and the risk-of-bias in non-randomized studies – for interventions (ROBINS-I) in cohort-type studies version to assess the risk of bias in the included studies. However, they do not provide enough detail on how these tools were used or the criteria used to assess the risk of bias. The authors should provide more information on the specific domains assessed by each tool and how the overall risk of bias was determined for each study. Additionally, the authors should report the inter-rater agreement for the risk of bias assessment.
Overall, the materials and methods section of this research article has several strengths, such as the use of a standardized review checklist, independent reviewers for study selection and data extraction, and assessment of the risk of bias in the included studies. However, there are also several areas where the methodology could be improved or clarified, such as providing more detail on the search strategy and study selection criteria, criteria for assessing relevance and quality of studies, and the risk-of-bias assessment. Additionally, the authors should provide a key or glossary to explain the simplified nomenclature.
Results:
Firstly, the authors should provide more information about the quality assessment of the included studies. While the article mentions that all studies were assessed for eligibility based on predefined criteria, there is no information provided on how the studies were evaluated for quality. It would be helpful if the authors could explain how they assessed the risk of bias in the included studies, and whether any studies were excluded due to poor quality.
Secondly, the article would benefit from a more detailed discussion of the findings. While the article presents a comprehensive overview of the included studies, there is limited discussion of the implications of the results or how they relate to previous research in this area. The authors should provide more context for their findings, and consider the limitations of the study and potential avenues for future research.
Finally, the article could be improved by including more information about the study population and intervention. While the article provides some information on the characteristics of the included studies, such as the number of participants and the age range, there is limited information provided on the specific populations and interventions evaluated in each study. Providing more information on the specific interventions and populations studied would allow readers to better understand the relevance and generalizability of the findings.
Discussion:
The discussion section could be more critical in evaluating the quality of the studies, including the potential biases and limitations. It would also be helpful to explore the reasons for the inconsistencies in the results of some studies, which could be due to differences in the study design or characteristics of the participants.
Moreover, the manuscript lacks clarity in presenting the results for different HPV types. The author needs to clearly indicate which HPV types were studied and the efficacy of the vaccine against each type. Finally, while the manuscript highlights the need for vaccination among high-risk populations such as MSM and PLWH, it fails to discuss the potential barriers to HPV vaccination among these groups, which could limit the effectiveness of the vaccine.
The manuscript contains many grammatical errors and awkward sentences. The authors should proofread the manuscript carefully and consider using a professional editor.
Author Response
REVIEWER 1
Comments and Suggestions for Authors
Overall, this research article presents a systematic review of the impact of the HPV vaccine in male disease, focusing on outcomes related to anal and genital HPV-related diseases. The authors report a detailed methodology for their literature search and selection process, including the number of retrieved papers, the number of studies assessed for eligibility, and the number of included studies. However, there are some areas where the article could be improved, as described below.
- Introduction:
The introduction provides a comprehensive overview of the prevalence of HPV-related diseases in males worldwide, highlighting the specific genotypes involved in different diseases. However, the introduction could benefit from a clearer and more structured presentation of the research question and objectives. The research question and objectives are briefly mentioned in the last sentence of the introduction, but a clearer statement of the research question at the outset would have better contextualized the subsequent discussion. Additionally, while the introduction provides information on the prevalence of HPV vaccination in males worldwide, it could benefit from further discussion on the barriers to vaccine uptake in males and the potential impact of increasing vaccine coverage in reducing the burden of HPV-related diseases. Overall, the introduction provides a good overview of the topic but could benefit from greater clarity and structure.
Reply:
Revised as suggested. |
Materials and methods:
Overall, the materials and methods section of this research article appears to be well-structured and clearly presented. However, there are several areas where the methodology could be improved or clarified, which I will discuss in detail below.
- Search Strategy and Study Selection Criteria:
The authors have stated that they followed the PRISMA checklist, which is a standard guideline for systematic reviews. However, they have not provided enough detail on the search strategy and study selection criteria to enable the reader to fully evaluate the quality of the review. For example, the authors state that they searched three electronic databases and ClinicalTrials.gov, but they do not specify the search terms used or the search date range for each database. It would also be helpful to know how many studies were identified from each database, how many were screened, and how many were included in the final analysis. Additionally, the authors should provide a flow diagram illustrating the study selection process. Furthermore, the authors state that they included studies that investigated the efficacy of vaccination against HPV-related diseases in males of any age and that had a control group of males who received placebo or were unvaccinated. However, it is unclear why the authors did not include studies that evaluated immunogenicity and infection incidence rates and prevalence, as these are important outcomes for assessing vaccine efficacy. The authors should provide a justification for this exclusion criterion.
Reply:
Revised as suggested. |
- Study Quality Assessment (Risk-of-bias):
The authors have stated that they used the Cochrane risk-of-bias tool for randomized trials (RoB2) and the risk-of-bias in non-randomized studies – for interventions (ROBINS-I) in cohort-type studies version to assess the risk of bias in the included studies. However, they do not provide enough detail on how these tools were used or the criteria used to assess the risk of bias. The authors should provide more information on the specific domains assessed by each tool and how the overall risk of bias was determined for each study. Additionally, the authors should report the inter-rater agreement for the risk of bias assessment.
Reply:
Revised as suggested. |
- Overall, the materials and methods section of this research article has several strengths, such as the use of a standardized review checklist, independent reviewers for study selection and data extraction, and assessment of the risk of bias in the included studies. However, there are also several areas where the methodology could be improved or clarified, such as providing more detail on the search strategy and study selection criteria, criteria for assessing relevance and quality of studies, and the risk-of-bias assessment. Additionally, the authors should provide a key or glossary to explain the simplified nomenclature.
Reply:
Revised as suggested. |
Results:
- Firstly, the authors should provide more information about the quality assessment of the included studies. While the article mentions that all studies were assessed for eligibility based on predefined criteria, there is no information provided on how the studies were evaluated for quality. It would be helpful if the authors could explain how they assessed the risk of bias in the included studies, and whether any studies were excluded due to poor quality.
Reply:
Revised as suggested. For the RoB2 tool, the overall risk-of-bias was determined using the Excel tool available on https://drive.google.com/uc?export=download&id=1malyRF_b-DgvAGHssrdt4N9R7Yhljmt0 For the ROBINS-I, the overall risk-of-bias was determined using the template available on https://drive.google.com/file/d/0B7IQVI0kum0kWldlU1BzRGxnclE/view?resourcekey=0-Glb26r8cajfG-bfA9PM6-w
|
- Secondly, the article would benefit from a more detailed discussion of the findings. While the article presents a comprehensive overview of the included studies, there is limited discussion of the implications of the results or how they relate to previous research in this area. The authors should provide more context for their findings, and consider the limitations of the study and potential avenues for future research.
Reply:
We carried out the discussion of the findings in the Discussion section of our paper. |
- Finally, the article could be improved by including more information about the study population and intervention. While the article provides some information on the characteristics of the included studies, such as the number of participants and the age range, there is limited information provided on the specific populations and interventions evaluated in each study. Providing more information on the specific interventions and populations studied would allow readers to better understand the relevance and generalizability of the findings.
Reply:
Revised as suggested. |
Discussion:
- The discussion section could be more critical in evaluating the quality of the studies, including the potential biases and limitations. It would also be helpful to explore the reasons for the inconsistencies in the results of some studies, which could be due to differences in the study design or characteristics of the participants.
Reply:
Revised as suggested. |
- Moreover, the manuscript lacks clarity in presenting the results for different HPV types. The author needs to clearly indicate which HPV types were studied and the efficacy of the vaccine against each type. Finally, while the manuscript highlights the need for vaccination among high-risk populations such as MSM and PLWH, it fails to discuss the potential barriers to HPV vaccination among these groups, which could limit the effectiveness of the vaccine.
Reply:
Revised. |
- Comments on the Quality of English Language
The manuscript contains many grammatical errors and awkward sentences. The authors should proofread the manuscript carefully and consider using a professional editor.
Reply:
Revised as suggested. |

Reviewer 2 Report
Introduction is well written, all 23 references are relevant.
Materials and Methods - Search strategy - exclusion criteria are described in lines 98 - 99, 101 - 107, 144 - 146, it is confusing. I suggest to simplify the text and to create a table clearly describing inclusion and exclusion criteria.
Study Quality Assessment - line 126 - correct typing error.
The same paragraph in the text - line 130 - 136 - it is acceptable to simplify the nomenclature LSIL as AIN GI, HGAIN and HSIL as AIN GII/III. The same LSIL as PIN GI and HGAIN and HSIL as PIN GII/III (discussed with histopathologist).
Results - well presented, well structured, tables are clear. In Table 1, typing error in Goldstone paper. In "Inclusion criteria", unify the spelling of "naive". If you use spelling "naïve", do it in the whole text.
Discussion - not well structured but acceptable. In the sentence line 344/345, omit "of course".
Author Response
REVIEWER 2
Comments and Suggestions for Authors
- Introduction is well written, all 23 references are relevant.
- Materials and Methods - Search strategy - exclusion criteria are described in lines 98 - 99, 101 - 107, 144 - 146, it is confusing. I suggest to simplify the text and to create a table clearly describing inclusion and exclusion criteria.
Reply:
Revised as suggested. |
- Study Quality Assessment - line 126 - correct typing error.
Reply:
Revised as suggested. |
- The same paragraph in the text - line 130 - 136 - it is acceptable to simplify the nomenclature LSIL as AIN GI, HGAIN and HSIL as AIN GII/III. The same LSIL as PIN GI and HGAIN and HSIL as PIN GII/III (discussed with histopathologist).
Reply:
We opted for the nomenclature we used due to it also being a validated nomenclature and being widely used in many articles, allowing us to simplify the reading experience. We also added a supplementary material simplifying the nomenclature (Supplementary Material 2). |
- Results - well presented, well structured, tables are clear. In Table 1, typing error in Goldstone paper. In "Inclusion criteria", unify the spelling of "naive". If you use spelling "naïve", do it in the whole text.
Reply:
Revised as suggested. |
- Discussion - not well structured but acceptable. In the sentence line 344/345, omit "of course".
Reply:
Revised as suggested. |

Reviewer 3 Report
1、 Could the results presented by the authors be represented by a more intuitive statistical graph?
2、 Did the authors stratified the analysis by type or dosage form of vaccine?
3、 The authors don't seem to have made it clear in the manuscript what the biggest impact of the HPV vaccine would be on men? This is the most important part of this study and also the most significant for the clinical work.
4、 This paper mainly focused on the safety of HPV vaccine, and there is a more important effectiveness result. I wonder whether the author has carried out relevant analysis.
Author Response
REVIEWER 3
Comments and Suggestions for Authors
1、 Could the results presented by the authors be represented by a more intuitive statistical graph?
Reply:
We did not use any statistical graphs to present the results since each paper have heterogenous data of population and assessed outcome (incident or recurrent disease). This type of presentation can lead to miss-comprehension of the by the reader. |
2、 Did the authors stratified the analysis by type or dosage form of vaccine?
Reply:
All retrieved papers used 4vHPV vaccine but one (which used 9vHPV vaccine). |
3、 The authors don't seem to have made it clear in the manuscript what the biggest impact of the HPV vaccine would be on men? This is the most important part of this study and also the most significant for the clinical work.
Reply:
Revised as suggested. |
4、 This paper mainly focused on the safety of HPV vaccine, and there is a more important effectiveness result. I wonder whether the author has carried out relevant analysis.
Reply:
Our focus was not safety precautions of vaccination and evaluation of adverse events of HPV vaccine. This manuscript aims to assess the efficacy of HPV vaccine on clinical manifestations of HPV-related diseases, and its prevention among men. |

Round 2
Reviewer 1 Report
There are a few points in the manuscript that could be improved for the clarity.
Authors should read the manuscript carefully and improve the sentences.
Few examples are below.
Line 31, instead of saying "are very prevalent," it would be better to say "have a high prevalence" or "are highly prevalent.
Line 57, instead of saying "The most frequent genotypes in anal diseases attributable to HPV are HPV-16 and HPV-18, with a prevalence of 70%," it would be clearer to say "The most frequent genotypes associated with anal diseases caused by HPV are HPV-16 and HPV-18, accounting for 70% of cases.
Line 60, instead of saying "People living with human immunodeficiency virus (HIV) infection (PLWH) have an estimated incidence rate for anal cancer of 35 per 100,000 person-years and a recurrence rate of high-grade anal intraepithelial neoplasia (HGAIN) of 50% within a year," it would be clearer to say "People living with human immunodeficiency virus (HIV) infection (PLWH) have an estimated incidence rate of 35 per 100,000 person-years for anal cancer and a recurrence rate of 50% for high-grade anal intraepithelial neoplasia (HGAIN) within a year.
Furthermore, discussion needs further improvements, as it lacks critical analysis and interpretation of the results. The authors should provide a comprehensive interpretation of the findings, discussing the implications and limitations of the included studies, as well as the potential reasons behind the observed variations in vaccine efficacy.
The language and clarity of the manuscript could be improved. Some sentences are lengthy and convoluted, making it difficult to follow the train of thought. Simplifying the language and organizing the information in a more coherent manner would enhance the readability of the manuscript.
The language and clarity of the manuscript could be improved. Some sentences are lengthy and convoluted, making it difficult to follow the train of thought. Simplifying the language and organizing the information in a more coherent manner would enhance the readability of the manuscript.
Author Response
Reply to Reviewer1
Authors should read the manuscript carefully and improve the sentences.
Reply:
Revised as suggested. |
Line 31, instead of saying "are very prevalent," it would be better to say "have a high prevalence" or "are highly prevalent.
Reply:
Revised as suggested. |
Line 57, instead of saying "The most frequent genotypes in anal diseases attributable to HPV are HPV-16 and HPV-18, with a prevalence of 70%," it would be clearer to say "The most frequent genotypes associated with anal diseases caused by HPV are HPV-16 and HPV-18, accounting for 70% of cases.
Reply:
Revised as suggested. |
Line 60, instead of saying "People living with human immunodeficiency virus (HIV) infection (PLWH) have an estimated incidence rate for anal cancer of 35 per 100,000 person-years and a recurrence rate of high-grade anal intraepithelial neoplasia (HGAIN) of 50% within a year," it would be clearer to say "People living with human immunodeficiency virus (HIV) infection (PLWH) have an estimated incidence rate of 35 per 100,000 person-years for anal cancer and a recurrence rate of 50% for high-grade anal intraepithelial neoplasia (HGAIN) within a year.
Reply:
Revised as suggested. |
Furthermore, discussion needs further improvements, as it lacks critical analysis and interpretation of the results. The authors should provide a comprehensive interpretation of the findings, discussing the implications and limitations of the included studies, as well as the potential reasons behind the observed variations in vaccine efficacy.
Reply
We have improved the discussion accordingly your suggestions.
The language and clarity of the manuscript could be improved. Some sentences are lengthy and convoluted, making it difficult to follow the train of thought. Simplifying the language and organizing the information in a more coherent manner would enhance the readability of the manuscript.
Reply:
We have reviewed the manuscript carefully and we have simplified the language.
Thank you very much .
Reviewer 3 Report
no more commets
Author Response
Reply to Reviewer.
no more comments.
Reply
Tank you very much.